# Performance Evaluation of Carbon Fiber Fabric-Reinforced Formaldehyde-Free High-Strength Plywood

**DOI:** 10.3390/polym16182637

**Published:** 2024-09-18

**Authors:** Yuanwu Wang, Qiheng Tang, Xueqi Chen, Xiaoxi Luo, Fenghao Zhang, Guanwu Zhou, Jie Zhang, Lei Zhang, Yuan Meng, Yiping Ren, Liang Chang, Wenjing Guo

**Affiliations:** Research Institute of Wood Industry, Chinese Academy of Forestry, No.1 Dongxiaofu, Haidian District, Beijing 10091, China

**Keywords:** plywood, maleic anhydride polyethylene, mechanical properties, reinforcement, mechanism

## Abstract

Plywood is lightweight, strong, and durable, making it a widely used material in building decoration and furniture areas. In this study, formaldehyde-free, high-strength plywood was prepared through the incorporation of carbon fiber fabrics (CFFs) as reinforcement layers and their bonding with maleic anhydride polyethylene (MAPE) films. Various tests were performed to assess the impact of the carbon fiber fabric positioning on the physical and mechanical properties of plywood, including tensile shear strength, flexural strength, water absorption, thickness swelling, and electro-thermal properties. The results revealed that the plywood with CFFs exhibited significantly higher mechanical properties than plywood without CFFs. Particularly, the addition of CFFs increased the tensile strength of the plywood by nearly 54.43%, regardless of the CFFs’ position. The symmetric placement of CFFs near the bottom and upper layers of the plywood resulted in a maximum modulus of rupture of 85.6 MPa. These findings were validated by numerical simulations. Scanning electron microscopy analysis of the plywood microstructures revealed that MAPE penetrated both the vessels and xylem of the wood veneers and the pores of the CFFs, thereby improving the mechanical properties of the plywood. Plywood reinforced with CFFs exhibited increased water absorption and thickness swelling after immersion. Additionally, the placement of CFFs influenced the electro-thermal properties of the plywood. Plywood with CFFs positioned near the bottom and upper surfaces exhibited superior thermal conductivity. Overall, this study presents a feasible method for developing high-performance, formaldehyde-free plywood and sustainable wood-based structural materials with potential applications in geothermal flooring.

## 1. Introduction

Plywood is a decorative wood-based panel made from wood veneers and adhesives. These veneers are arranged in an odd number of layers (including 3, 5, and 7) through a cross-lamination process. Plywood is widely used in construction, furniture, decking, etc. [1,2,3,4]. Currently, plywood is mainly produced using formaldehyde-based adhesives, such as urea–formaldehyde resins, phenol–formaldehyde resins, and melamine–urea–formaldehyde resins. The use of these adhesives results in formaldehyde emissions during the production of plywood, posing significant hazards to both the environment and human health [5,6]. With increasing environmental awareness, concerns regarding human health have received significant attention. Numerous researchers have progressively developed plywood using formaldehyde-free adhesives. Among these new adhesives, high-density polyethylene films have attracted significant interest from both industry and academia [7,8,9,10].

With the introduction of carbon neutrality policies, numerous researchers have investigated the use of plywood in semi-structural and structural applications. However, owing to the scarcity of forest resources in China, the raw wood veneers used for plywood mainly originate from fast-growing species, such as poplar and eucalyptus [11]. These woods often contain many knots and exhibit lower and more variable mechanical properties. To address these challenges, previous studies have explored various methods to enhance the mechanical properties of plywood, including selecting different wood species, refining veneer processing techniques, and incorporating various types of resins. Bekhta P. et al. investigated the effects of wood species on plywood properties. The results revealed that softwoods exhibited significantly lower modulus of rupture (MOR), modulus of elasticity (MOE), and bonding strength than hardwoods. Particularly, beech plywood exhibited the highest bending strength properties, followed by birch plywood and spruce plywood [12]. Additionally, Temiz A. et al. found that treating veneer with plasma using O_2_ enhanced the bonding strength of beech plywood compared with untreated wood [13]. Moreover, Bekhta P. et al. found that plywood treated with citric acid exhibited superior shear and bending strength and reduced water absorption (WA) and thickness swelling (TS). The results revealed that birch and black alder plywood panels had the highest bending strength properties, followed by grey alder and aspen [14]. Similarly, Kallakas H. et al. reported that birch and black alder plywood panels exhibited the highest bending strength properties, followed by grey alder and aspen. However, with appropriate lay-up schemes, these wood species can be effectively used in the veneer-based product industry [15]. Furthermore, Chen X. et al. investigated the effect of different adhesives on the mechanical properties of plywood. The results revealed that the modified starch adhesive-based plywood at pH 4.50–5.50 exhibited higher wet shear strength than the unmodified starch adhesive-based plywood. This improvement can be attributed to the ability of the modified starch adhesive to penetrate the bonding interface of the plywood [16]. However, these methods only slightly enhanced the mechanical properties.

Currently, man-made fiber reinforcement is a highly effective method for improving the mechanical properties of wood-based panels. Research has shown that incorporating fibers (such as carbon, glass, and basalt fiber) into wood-based panels, such as glulam and laminated veneer lumber (LVL), can significantly improve their mechanical properties [17,18,19,20,21]. Notably, carbon fiber is widely used in structural applications owing to its low density, high tensile strength, stiffness, and chemical resistance [22,23]. In recent years, carbon fiber fabrics (CFFs) have been commonly used to enforce LVL composite materials. Bakalarz M. et al. found that reinforcing LVL with carbon fabric sheets improved its bending strength by 30% [24]. Moreover, Rescalvo et al. enhanced shear and compression strength by reinforcing LVL with both carbon and basalt fibers [25]. Mercimek Ö. et al. reinforced glulam wooden beams with carbon fiber-reinforced polymer. The study indicated that the use of carbon fiber strips significantly increased the load-bearing capacity of the beams and had a highly positive effect on their overall load–displacement behavior [26]. Additionally, Gallego et al. reinforced LVL panels with carbon and basalt fibers to improve their bending properties. However, the reinforcement fibers were placed between the last and penultimate veneers on each side of the board [27].

Nevertheless, the use of high-density polyethylene as adhesives and CFFs as reinforcement in plywood has not been extensively investigated. Therefore, this study aimed to use CFFs to produce high-strength plywood. Because carbon fiber surfaces are non-polar and non-reactive and wood veneer surfaces are polar, maleic anhydride polyethylene (MAPE) was used as an adhesive to improve interfacial adhesion.

In this study, formaldehyde-free, high-strength plywood was prepared using CFFs as reinforcement with different veneer arrangements bonded with MAPE adhesives. The mechanical, water-resistant, electro-thermal properties of plywood were characterized. Additionally, the microstructures of the plywood cross-sections were analyzed via scanning electron microscopy (SEM) to elucidate the underlying mechanisms. This study presents a feasible method for developing high-performance, formaldehyde-free plywood and sustainable wood-based structural materials, with potential applications in geothermal flooring. The obtained plywood exhibits superior mechanical properties, is formaldehyde-free, and presented intriguing thermoelectric conversion capabilities. The CFF-reinforced plywood is simple, rapid, and feasible to process, which is expected to realize industrialized production. All the raw materials used are sustainable and environmentally friendly, without complex chemical treatment. Therefore, a simple, sustainable, and processable CFF-reinforced plywood would optimize the wood-based panel configuration and effectively enhance the social benefits.

## 2. Materials and Methods

### 2.1. Materials

*Eucalyptus* veneers (*Eucalyptus* spp.) with dimensions of 300 × 300 × 1.7 mm and an average moisture content of ~5% were provided by Xiayi Jinzhan Wood Industry Co., Ltd. (Shangqiu, China). CFFs, produced by Zhongfu Shenying Carbon Fiber Co. Ltd. (Lianyungang, China), consisted of plain weave type-3k yarn. The thickness of CFFs is 2 mm, used bidirectionally. According to the manufacturer, these fibers exhibited a tensile strength of 3500 MPa, a MOE of 230 GPa, and an average density of 1.77 g·cm^−3^. MAPE films were purchased from Fujian Qingxin Technology Co., Ltd. (Fuzhou, China), China.

### 2.2. Production of Plywood

Figure 1 shows the configuration of four types of plywood. The first type was the control sample composed of seven veneers bonded with the MAPE film adhesive. The other three types included plywood with two layers of CFFs in different locations. These plywood types were denoted as CFFs/PI, CFFs/PII, and CFFs/PIII. After proper preparation of the materials, the laminate underwent hot pressing and cold pressing processes. During the hot pressing phase, the machine was heated to 160 °C, and a pressure of 1.0 MPa was applied to the laminate for 10 min. After hot pressing, the laminate was immediately transferred to a cold press machine set at 20 °C and subjected to the same pressure and duration as used in the hot pressing process (Figure 1). The cold pressing accelerated the cooling process of the MAPE film adhesive. Finally, formaldehyde-free, high-strength plywood was obtained, and five specimens were fabricated for each series. Density is a very important property for characterizing panels, and the density of all samples was 0.69 ± 0.03 g/cm^3^.

### 2.3. Characterization

#### 2.3.1. Mechanical Properties

A tensile strength test was conducted on both the control and reinforced plywood according to GB/T 1447-2005 [28]. A crosshead speed of 5 mm·min^−1^ was used. The flexure properties were characterized according to GB/T 17657-2022 [29], and the samples were 250 mm × 50 mm × 10 mm (length × width × thickness). The test was performed at a crosshead speed of 2 mm·min^−1^ with a span of 64 mm. At least five specimens were tested for each series. The original dimensions of the produced panels were 30 cm × 30 cm. For each treatment, 3 panels were manufactured. A total of five samples were made for each physical and mechanical test performed, from which the mean value was derived for analysis.

#### 2.3.2. SEM Analysis

The micromorphology of the cross-sections of plywood with and without CFFs was analyzed using scanning electron microscopy (SU4800, HITACHI, Toyko, Japan). The cross-sections from the samples were obtained by freezing microtome. The micromorphology of the dried CFF-reinforced plywood (103 °C, 2 h) was analyzed via scanning electron microscope (SEM). To better characterize the sample micro-morphology, all the SEM samples were metalized. Specifically, the SEM samples were cut into 1 × 1 × 0.5 cm^3^ blocks and sputtered with gold for observation. The experiment was performed by a Hitachi S-4800 SEM (Hitachi, Ltd., Tokyo, Japan) with a secondary electron (SE) mode at an accelerating voltage of 5 kV.

#### 2.3.3. Finite Element Simulation Analysis

The simulation of mechanical properties was carried out by finite element software, where eucalyptus wood is considered an orthotropic material, while CFRP, support rollers, and loading rollers are considered isotropic materials. The contact properties between each layer adopt adhesive behavior, with the damage criterion being the maximum nominal stress criterion and the damage evolution type being linear displacement. The contact type between the support roller, loading roller, and board is surface-to-surface contact, and rigid constraints are applied to the support roller and loading roller. The boundary condition type of the support roller is “symmetric/anti symmetric/completely fixed”, and the boundary condition type of the loading roller is “displacement/rotation angle”. The unit type is set to eight-node quadratic tetrahedral unit (C3D8R) to reduce running time and improve accuracy.

#### 2.3.4. Water Resistance

Thickness swelling (*TS*) and water absorption (*WA*) were measured according to GB/T 17657-2022, with each specimen measuring 50 mm in length and 50 mm in width. Samples were soaked in water at room temperature for different times. A total of 5 specimens were used for each test. Thickness swelling was calculated from the same samples with the following Equation (1):(1)$$TS\left(\%\right)=\frac{h_{2}-h_{1}}{h_{1}}\times 100\%$$
where *TS* is the thickness swelling (%), *h*_1_ is the thickness of the sample before soaking, and *h*_2_ is the thickness of the sample after soaking.

Water absorption was calculated from the following Equation (2):(2)$$WA(\%)\;=\frac{m_{2}-m_{1}}{m_{1}}\times 100\%$$
where *WA* is water absorption (%), *m*_1_ is the weight of the sample before soaking, and *m*_2_ is the weight of the sample after soaking.

#### 2.3.5. Electrothermal Properties

The electrothermal conversion test was conducted on the reinforced plywood. The samples had dimensions of 50 mm × 50 mm (length × width). The Decagon KD2 PRO equipment (UNI-TERND Technology (China) Co. Ltd., UT301A, Dongguan, China) was applied to assess the electrothermal properties of the samples. The CFF-reinforced plywood was processed into 5 × 5 × 1 cm^3^ blocks. Subsequently, copper wires were connected to carbon fiber layers of samples. Finally, the samples were directly connected to an adjustable direct current (DC) power. Then, a Decagon KD2 PRO thermocouple was used to monitor the temperature change in real time.

## 3. Results

### 3.1. Tensile Property Analysis

Figure 2a shows the tensile strength of plywood with and without CFF reinforcement. The control sample exhibited a flexural strength of 36.41 MPa. The CFF-reinforced plywood exhibited significantly higher tensile strength, owing to the superior tensile properties of CFFs compared with the veneer. Particularly, the tensile strength of CFF/PI increased by 54.43% compared with the control wood. However, no significant differences were observed in the tensile strength of plywood containing CFFs placed in different locations within the multilayer structure, indicating that the position of CFFs did not affect the tensile strength. To investigate the extent of the influence of different locations on tensile property, it is necessary to carry out an analysis of variance using ANOVA. In statistical analysis, the different Pr value indicates different meanings. When the Pr < 0.01, this indicates that the set of data features an extremely significant effect, and when 0.01 < Pr < 0.05, this indicates that the set of data has a significant effect. However, when the Pr > 0.05, this indicates that the set of data has no significant effect. The ANOVA analysis results are presented in Table 1. It can be seen that the locations of CFFs have extremely significant effects on the tensile property (Pr < 0.01) in statistical analysis. Moreover, the differences between the plywood samples without and with CFFs were also analyzed by *t*-test. The analysis result is shown in Figure 2a and Table 2. The same letter over the column indicates that the means of the paired groups have no significant difference (Pr > 0.05), and a vs. b means extremely significant difference (Pr < 0.01). According to the criteria, it can be seen that the tensile properties of plywood without CFFs and with CFFs differ significantly, but the tensile properties of plywood with CFFs do not significantly differ from each other.

The tensile shear strengths of the plywood samples are shown in Figure 2b, which gives clear insight into the bonding strength of MAPE with wood veneers and CFFs. It can be observed that the tensile shear strength of MAPE with wood veneers and CFFs are 1.25 MPa and 0.94 MPa, respectively. After the plywood was immersed in boiling water for 4 h, dried at 60 ± 3 °C for 16 h, and re-immersed in boiling water for 4 h, the tensile shear strength of MAPE reinforced with wood veneers and CFFs decreased to 0.87 and 0.62 MPa, respectively. However, the tensile shear strength between MAPE and wood veneers still met the requirements for type I grade plywood according to GB/T 9846.3 (2015) [30]. This indicates that the MAPE film exhibited good bonding strength. To investigate the influence of the boiling process on the tensile shear strengths of wood–wood and wood–CFFs, the relative values were also analyzed by *t*-test, and the results are shown in Table 3. It can be seen that the boiling process has no significant effect on the tensile shear property of wood–wood (Pr > 0.05) in statistical analysis. However, the boiling process had a significant effect on the tensile shear strength of wood–CFFs (0.01 < Pr < 0.05).

Figure 2 shows the common failure modes of plywood with different reinforcement configurations identified from the experimental tests. The specimens exhibited both tensile failure and shear delamination across all groups (Figure 2d–g). Particularly, tensile failure occurred in both transverse and longitudinal veneers, while shear delamination was observed between the veneers. These failures occurred simultaneously, likely owing to the symmetrical structure of the plywood, which distributed stress across the entire specimen [31].

### 3.2. Flexural Property Analysis

Figure 3 shows the flexural performances of the plywood samples. The control sample exhibited a mean MOR and MOE of 60.10 and 9670 MPa, respectively. The addition of CFFs significantly increased the MOR of the plywood, regardless of the CFFs’ placement. Notably, CFFs positioned near the external veneers provided the most significant reinforcement effect. CFFs/PI, CFFs/PII, and CFFs/PIII exhibited flexural strengths of 71.03, 82.90, and 85.60 MPa, respectively, which exceeded those of the control sample by 18.10%, 37.91%, and 42.42%, respectively (Figure 3). Additionally, CFFs/PI, CFFs/PII, and CFFs/PIII exhibited MOE values of 9160, 9510, and 9040 MPa, respectively, which were nearly the same as those of the control samples. The symmetrical positioning of two CFFs on the bottom and upper adhesive lines achieved the highest reinforcement effect. This suggests that the positioning of the carbon fiber sheet in the sandwich structure had a greater impact on flexural performance than on tensile performance. The reinforcement of plywood with carbon fiber in the face layer was significantly more effective than its placement in the core layer of the material. During bending strength tests, tensile forces developed near the bottom surface of the test specimen and extended toward the support points as the test progressed. Under this tensile force, the plywood failed at the bottom layer. However, owing to the significantly higher tensile strength of the carbon fiber sheet than the veneer [32,33], the CFFs/PIII composite exhibited stronger flexural strength.

To investigate the influence of the locations of CFFs on the flexural properties of the resulting plywood, it was necessary to carry out an analysis of variance, the results of which are presented in Table 4. The ANOVA analysis result indicates that the different locations of CFFs have extremely significant effects on MOR (Pr < 0.01) in statistical analysis. However, the different locations of CFFs do not have significant effects on MOE (Pr > 0.05). Further, the differences of MOR and MOE among the plywood without and with CFFs were also analyzed by *t*-test. The analysis results are shown in Figure 3a,b, and the relative data are shown in Table 5 and Table 6. The same letter over the column indicates that the means of the paired groups have no significant difference (Pr > 0.05), and a vs. b indicates an extremely significant difference (Pr < 0.01). According to the criteria, it can be seen that the MOR between plywood without CFFs and with CFFs differs significantly, but the MOE among plywood with CFFs does not significantly differ.

Figure 4 shows the stress distribution patterns of plywood with and without CFF reinforcement. During the bending test, the specimens exhibited compressive stress on the upper side and tensile stress on the bottom side of the specimen (Figure 4). Plywood without CFFs fractured under smaller loads. However, plywood with CFFs positioned in the outer layer of the veneer, such as CFFs/PIII composites, could withstand larger loads. Overall, the finite element model numerical simulations in this study were consistent with the experimental results, confirming the accuracy and effectiveness of the experimental findings.

### 3.3. Microstructure Analysis

To investigate the bonding mechanism of the MAPE, the microstructures at the cross-section between the veneers and the MAPE were characterized via SEM. Figure 5a–f show the SEM images of the samples. Notably, the images reveal a distinct layered structure in the plywood. The CFFs and veneers were tightly bonded by MAPE films (Figure 5a,d), indicating good compatibility between the MAPE and both veneers and CFFs. The raw wood veneers exhibited a complex porous structure, with MAPE penetrating the vessels and xylem of the veneers (Figure 5b). This penetration facilitated mechanical interlocking [34], thereby enhancing the bonding properties. Moreover, during hot pressing, the CFFs were filled with MAPE (Figure 5e). The combination of the low porosity structure and the high strength of CFFs significantly enhanced the mechanical properties of the plywood.

The compatibility between MAPE with both veneers and CFFs was also characterized by SEM with larger magnification. The results show that the interfacial adhesion between the MAPE and the veneer was highly compatible, with no discernible gaps at the interface (Figure 5c). However, for the interfacial compatibility between CFFs and MAPE (Figure 5f), it can be seen that due to the low addition of MAPE, MAPE merely penetrated into the CFFs to realize the interlocking of MAPE and CFFs.

### 3.4. Water-Resistant Property Analysis

Figure 6 shows the *TS* and *WA* values of plywood after water immersion at room temperature for different durations. Initially, all samples exhibited uniform thickness. As the soaking time increased, *TS* values significantly increased. After 24 h of immersion, *TS* values reached a plateau, indicating that the plywood reached a near-saturation state. Notably, plywood with CFFs exhibited higher TS than plywood without CFFs despite having similar initial thickness. This suggests that plywood with CFFs had a more densified structure during hot pressing, which expanded the dimensions of the panels, leading to higher swelling. Moreover, plywood with CFFs exhibited nearly uniform *TS*, indicating that the position of CFFs did not affect *TS*. Both the plywood without and with CFFs exhibited equal WA at 12 h, respectively. However, after 12 h, the plywood with CFFs had lower *WA* than the control sample, as the CFF-reinforced samples were heavier. Despite having the same thickness and veneer layers, the CFFs/PI, CFFs/PII, CFFs/PIII, and control samples absorbed the same amount of water. Nonetheless, the control sample exhibited a lighter weight, resulting in higher *WA*.

### 3.5. Electrothermal Property Analysis

CFF-reinforced plywood has the potential to be developed into a novel electrothermal device because of the excellent electrical conductivity of carbon fiber. Figure 7a,b show the temperature variations and schematic of the thermal conductivity test for plywood samples with CFFs. The schematic diagram of the composite as an electrothermal conversion device is demonstrated in Figure 7a. According to Joule’s Law, when current flows through a resistor, the electrical energy is converted into thermal energy, which can be represented by the following formula:(3)$$Q=I^{2}Rt=\frac{U^{2}}{R}t$$
where *Q* represents the generated thermal energy, *U* represents the applied voltage, *R* represents the resistance, and *t* stands for the working time. The Joule heating effect theory shows that the thermal energy transformation of the composites has a positive correlation with the square value of the applied voltage (*U*^2^) at different working times. This means that only a low voltage is required to rapidly heat carbon fiber layers. Natural eucalyptus plywood exhibited poor thermal conductivity with a slow temperature increase. The addition of CFFs significantly enhanced the thermal conductivity of the composites (Figure 7c). Particularly, the position of the carbon fiber in the composites slightly influenced their thermal conductivity. As the carbon fiber was positioned closer to the slab surface, the thermal conductivity of the composite gradually increased. After 20 min, the temperature plateaued. Moreover, the addition of carbon fibers, which are conductive materials, significantly reduced the electrical resistivity of the plywood. Compared with the CFFs/PI, when the temperature stabilized, the thermal conductivity of the CFFs/PIII increased by 9.72%. This suggests that the composite structure of the plywood reinforced with CFFs enhanced heat transfer, making it suitable for applications in electric-heating flooring areas.

## 4. Conclusions

This study investigated the effects of CFF reinforcement on the mechanical and physical properties of plywood made from eucalyptus veneers and MAPE film adhesives. CFFs reinforcement of the plywood not only enhanced the mechanical properties of the composite materials but also provided it with excellent electrothermal conversion properties. The utilization of MAPE film as a formaldehyde-free adhesive offers a dual advantage in the recycling of plastics and material conservation, which is expected to bring great economic and ecological benefits (e.g., by reducing white pollution). The results revealed that the addition of CFFs significantly increased the tensile strength of the plywood by ~54.43%, with consistent improvements across different CFF positions. However, the enhancement in the flexural strength of plywood reinforced with CFFs was significantly influenced by the specific positions of the CFFs. Particularly, the placement of CFFs near the outer veneers significantly increased MOR and MOE. Plywood with CFFs positioned near the surface veneer exhibited the highest MOR, representing a 42.42% increase. SEM analysis revealed that MAPE effectively served as an adhesive in plywood containing CFFs. Additionally, a significant amount of MAPE penetrated the cell lumens and pores of both wood veneers and CFFs, forming some interfacial bonds that improved the mechanical properties of the plywood. Regarding water resistance, both the plywood with and without CFFs exhibited a rapid increase in *WA* and *TS* during the first 24 h of immersion, followed by a plateau phase. The addition of CFFs increased the *TS* of plywood after water immersion but reduced its *WA*. Moreover, incorporating CFFs into CFFs/PIII enhanced its thermal conductivity, resulting in a 9.72% increase compared with the control sample. This indicates that CFF reinforcement significantly enhanced the mechanical properties, light weight, and strength of the plywood. Therefore, reinforced plywood can serve as a viable alternative to high-quality solid wood and other high-strength materials. Currently, plywood has been applied in the decorative interiors of China High-Speed Rail. Through our research, CFF-reinforced plywood has stronger mechanical properties and is expected to be applied in some structural material components of China High-Speed Rail, which broadens the scope of its application. Moreover, CFF-reinforced plywood also has good electrothermal properties, which is expected to be applied to the electric floor of China High-Speed Rail.

## Figures and Tables

**Figure 1 polymers-16-02637-f001:**
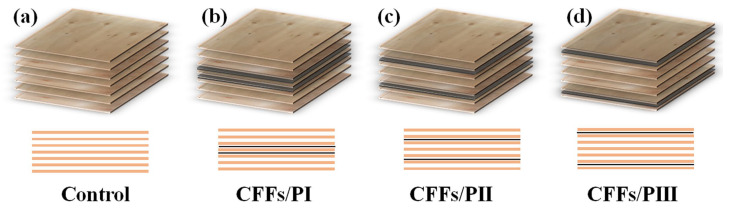
Configurations of plywood types: (**a**) control plywood; (**b**) CFFs/PI; (**c**) CFFs/PII; (**d**) CFFs/PIII.

**Figure 2 polymers-16-02637-f002:**
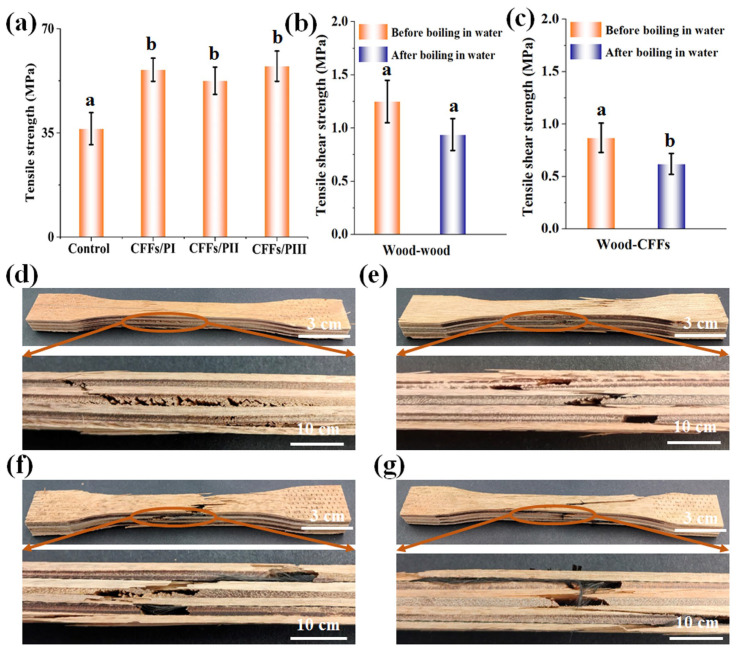
Effect of CFFs positions on plywood properties: (**a**) tensile strength; (**b**,**c**) tensile shear strength; (**d**) tensile failure modes in control plywood; (**e**) tensile failure modes in CFFs/PI; (**f**) tensile failure modes in CFFs/PII; (**g**) tensile failure modes in CFFs/PIII. ( Note: Data were analyzed by one-way ANOVA; the same letter over the column indicates that the means of the paired groups do not significantly differ; a vs b means significantly differ).

**Figure 3 polymers-16-02637-f003:**
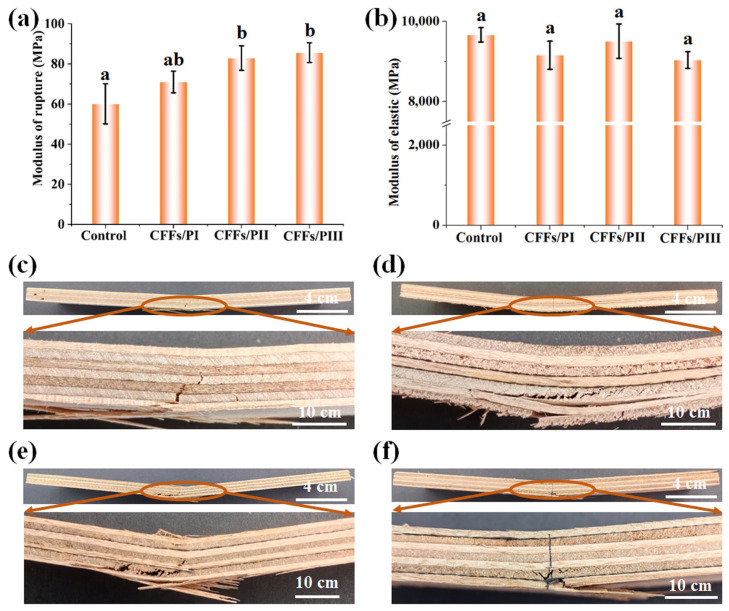
Effect of CFF positions on MOR (**a**) and MOE (**b**) of plywood. (**c**) bending failure modes in control plywood; (**d**) bending failure modes in CFFs/PI; (**e**) bending failure modes in CFFs/PII; (**f**) bending failure modes in CFFs/PIII. (Note: Data were analyzed by one-way ANOVA; the same letter over the column indicates that the means of the paired groups do not significantly differ; ab vs. a or b also means not significantly differ; a vs. b means significantly differ).

**Figure 4 polymers-16-02637-f004:**
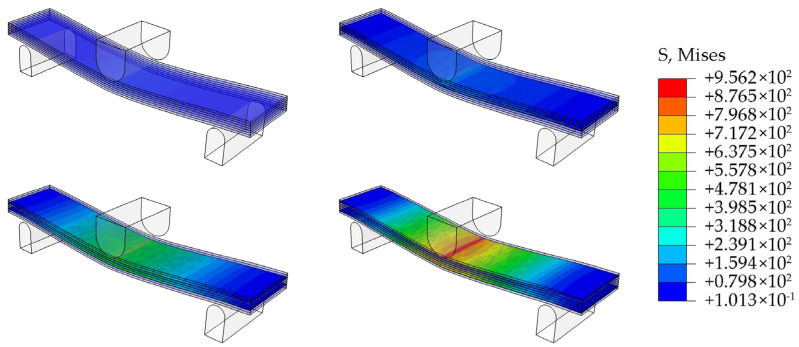
Stress contour of plywood with and without CFFs during the bending test.

**Figure 5 polymers-16-02637-f005:**
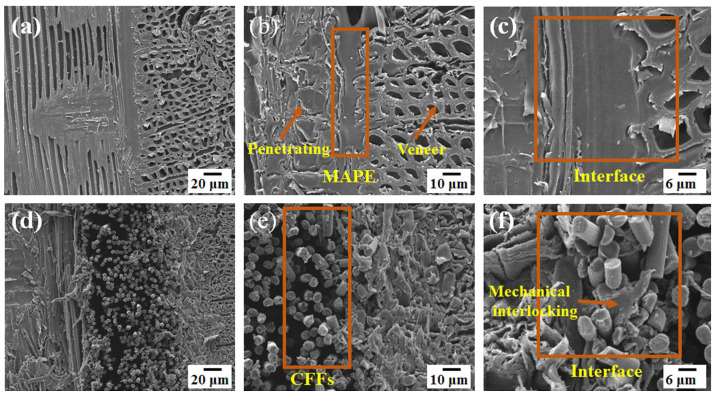
SEM micrographs of plywood cross sections: (**a**–**c**) control group (×200, ×500, ×800 magnification, respectively); (**d**–**f**) CFFs/PI (×200, ×500, ×800 magnification, respectively).

**Figure 6 polymers-16-02637-f006:**
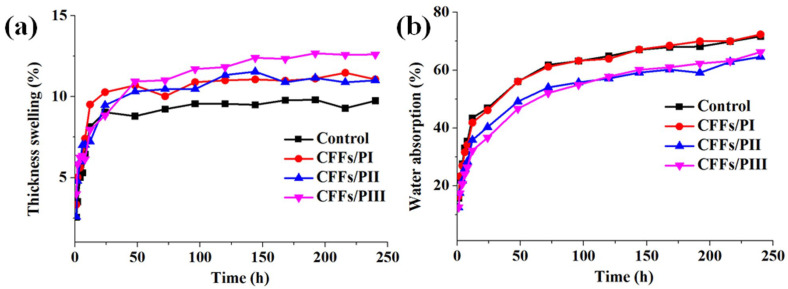
Variations in TS (**a**) and WA (**b**) of plywood with and without CFFs.

**Figure 7 polymers-16-02637-f007:**
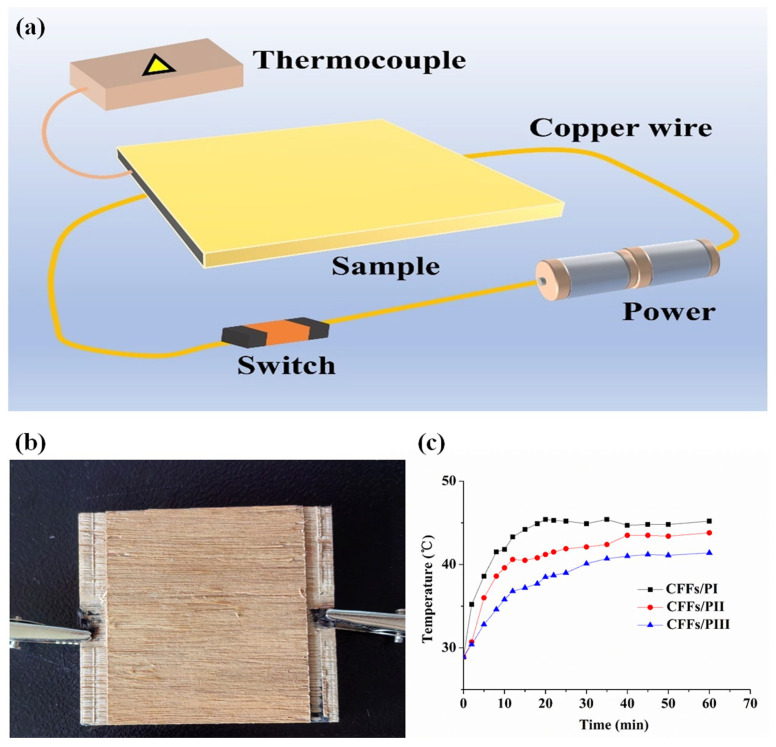
Electrothermal Property of plywood with CFFs in different locations. (**a**) schematic of the thermal conductivity test for plywood samples with CFFs; (**b**) image of testing process; (**c**) temperature variations curves for plywood samples with CFFs.

**Table 1 polymers-16-02637-t001:** ANOVA analysis for the tensile strength of plywoods.

Type	Squares	df	Mean Square	F Value	Pr
Tensile property	1029.27	20	282.39	12.41	0.002

**Table 2 polymers-16-02637-t002:** *t*-test analysis for the tensile strength of plywoods.

Type	Mean Difference	q	Pr
Control/CFFsPI	19.8	7.19	0.004
Control/CFFsPII	16.1	5.85	0.008
Control/CFFsPIII	−3.7	1.34	0.0029
CFFsPI/CFFsPII	20.97	7.61	0.7801
CFFsPI/CFFsPIII	1.17	0.42	0.9899
CFFsPII/CFFsPIII	4.87	1.77	0.6196

**Table 3 polymers-16-02637-t003:** *t*-test analysis for the tensile shear strength.

Type	Mean Difference	q	Pr
Wood–wood before boiling in water/ Wood–wood after boiling in water	−0.38	3.81	0.0543
Wood–CFFs before boiling in water/ Wood–CFFs after boiling in water	−0.32	4.34	0.0371

**Table 4 polymers-16-02637-t004:** ANOVA analysis for the flexural properties of plywood.

Type	Squares	df	Mean Square	F Value	Pr
MOR	1618.98	20	412.74	8.67	0.007
MOE	1,548,100	20	26,010	2.71	0.1152

**Table 5 polymers-16-02637-t005:** *t*-test analysis for the MOR of plywoods.

Type	Mean Difference	q	Pr
Control/CFFsPI	10.9	2.73	0.2870
Control/CFFsPII	22.8	5.72	0.0156
Control/CFFsPIII	25.5	6.40	0.008
CFFsPI/CFFsPII	11.9	2.99	0.2279
CFFsPI/CFFsPIII	14.6	3.67	0.1182
CFFsPII/CFFsPIII	2.7	0.68	0.9616

**Table 6 polymers-16-02637-t006:** *t*-test analysis for the MOE of plywoods.

Type	Mean Difference	q	Pr
Control/CFFsPI	−510	2.85	0.2586
Control/CFFsPII	−160	0.89	0.9186
Control/CFFsPIII	−630	3.52	0.1362
CFFsPI/CFFsPII	350	1.96	0.5417
CFFsPI/CFFsPIII	−120	0.67	0.9627
CFFsPII/CFFsPIII	−470	2.63	0.3152

## Data Availability

The original contributions presented in the study are included in the article, further inquiries can be directed to the corresponding author.

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
