# Peer review of "Performance Evaluation of Carbon Fiber Fabric-Reinforced Formaldehyde-Free High-Strength Plywood"

_polymers, 2024, doi:10.3390/polym16182637_

Round 1

Reviewer 1 Report

Comments and Suggestions for Authors

Comments

My comments for the manuscript entitled, ‘Performance evaluation of carbon fiber fabrics-reinforced formaldehyde-free high-strength plywood’ are given below,

1.     For the CFF reinforced plywood, there is an increase and decrease in the tensile strength from CFFs/PI to CFFs/PIII. Comment on this.

2.     Citation for Figures 2c-f was wrongly mentioned as Figure 5. Correction is required.

3.      Include the SEM images of CFFs/PII and CFFs/PIII to understand in detail about the compatibility between MAPE with both veneers and CFFs.

4.     The manuscript deals with more of the material property, and not about polymers. Therefore, it will be suitable for some other journals related to material property.

Author Response

Dear editor and reviewers:

Thank you for your letter and the reviewers’ comments concerning our manuscript (Manuscript ID: polymers-3180395) entitled “Performance evaluation of carbon fiber fabrics-reinforced formaldehyde-free high-strength plywood”. Those comments are not only valuable and helpful for revising and improving our manuscript, also have the important guiding significance to our researches. We have studied the comments carefully and made relevant corrections which we hope to meet your approval. The main corrections in this manuscript and the responds to the reviewer’s comments are as following:

Reviewer 1:

Comments and Suggestions for Authors

Comments

My comments for the manuscript entitled, ‘Performance evaluation of carbon fiber fabrics-reinforced formaldehyde-free high-strength plywood’ are given below,

Response: Thank you very much for the reviewer's recognition of our manuscript. Your suggestion not only helps us to correct the mistakes in our manuscript, but also expanded the depth of our research. Your recognition provides us with tremendous motivation to continue our further research.

  1. For the CFF reinforced plywood, there is an increase and decrease in the tensile strength from CFFs/PI to CFFs/PIII. Comment on this.

Response: Thank you for this comment. For the tensile strength, all the CFFs-reinforced plywood exhibited higher tensile strength compared with the control (without CFFs). As for the reviewer's comment about an increase and decrease in the tensile strength from CFFs/PI to CFFs/PIII, my understanding is that the variability in the tensile strength enhancement of the CFF reinforced plywoods. This may be attributed to the fact that the CFFs are placed at different interlaminar regions of the plywood. The tensile properties data of the specimens exhibit only marginal variations. When considering the error bars, the discrepancies in tensile properties of CFFs/PI, CFFs/PII, and CFFs/PIII are found to be statistically insignificant. The corresponding analysis and discussion have been added into the section 3.1.

  1. Citation for Figures 2c-f was wrongly mentioned as Figure 5. Correction is required.

Response: Thank you very much for this suggestion. The error in the section 3.1 has been corrected.

  1. Include the SEM images of CFFs/PII and CFFs/PIII to understand in detail about the compatibility between MAPE with both veneers and CFFs.

Response: Thank you for this comment. We used scanning electron microscopy with larger magnification to understand in detail about the compatibility between MAPE with both veneers and CFFs. The results show that the interfacial adhesion between the MAPE and the veneer was highly compatible, with no discernible gaps at the interface (Figure 5c). However, for the interfacial compatibility between CFFs and MAPE (Figure 5f), it can be seen that due to the low addition of MAPE, MAPE merely penetrated into the CFFs to realize the interlocking of MAPE and CFFs, but a large aera of interfaces were not formed between MAPE and CFFs, thus making it difficult to evaluate the interfacial compatibility.

  1. The manuscript deals with more of the material property, and not about polymers. Therefore, it will be suitable for some other journals related to material property.

Response: Thank you very much for this comment. In this study, we used MAPE polymer resin as an adhesive to prepare high-strength plywood. Our research was primarily directed towards evaluating the performance characteristics of this composite plywood. As the reviewer said, the main topic of this article is to achieve use CFFs as reinforcement to produce high-strength plywood. Therefore, we mainly focus on the material properties of CFFs reinforced plywood. And we have explored few contents about polymers within the limited research space. The reviewer's comment is very valuable and helpful to us, and our future research will explore more about the polymers analysis of the composite. Thank you very much for the reviewer's comment again.

Reviewer 2 Report

Comments and Suggestions for Authors

Abstract
(Line 28) - It's crucial to avoid repeating words from the title in the keywords. Let's replace the words: "high-strength" and "carbon fiber fabrics" to ensure the keywords are unique and effective.

Introduction
What is new about using this material? What benefits can it provide to industry, the environment and society?

Material and Methods
Was the carbon fiber fabric used unidirectionally or bidirectionally?
What is the thickness of the carbon fiber fabric?
What criteria were established for choosing the layer configurations between plywood and CFFs?
What is the size of the panels produced? How many panels were manufactured for each treatment evaluated? How many samples were made for each physical and mechanical test performed? Why were density tests not performed on the materials? Density is a very important property for characterizing panels. I suggest adding it. The authors should include the physical and mechanical tests in different topics. Water absorption and Thickness swelling are physical and not mechanical analyses.
How were the samples prepared for the SEM? Were the samples metalized, and what material was used for the metallization? Provide more details on the analysis procedure.
Detail the electrothermal conversion test. How were the samples prepared? What settings were used in the analyses? Did the analyses follow any standard or methodology?
This is important information that needs to be added. Why was no statistical test applied to the results obtained? It is important to check whether or not there is significance between the treatments evaluated.

Results and Discussion
To determine whether the values ​​indicated by the results showed significant differences between them, normalization, homogeneity, and variance analysis tests, followed by some mean test that applies to the analyses performed, must be performed. The fact that the values ​​are higher or lower is not enough to state that a significant difference occurred. For this, statistical tests are necessary. I suggest applying statistical tests to the results obtained.
(Line 176) – Figure 5 or Figure 2? Perform correction.
Add scale to Figures 2c, 2d, 2e and 2f. What are the modulus of elasticity values during the tensile strength tests? Why did the authors not choose to display the results based on the stress x deformation curve? Standardize all results to two decimal places after the point. Add scale to Figures 3c, 3d, 3e and 3f. In Figure 4, does each image represent an analyzed material? If so, it is important to identify them in the Figure. Are the images in Figure 4 simulations of the analyzed material? What software was used? How was the material assigned? What are the characteristics of the mesh created for the analyzed materials? What failure criterion was used? The experimental procedures must be presented in the methodology if a simulation is performed.
Markings should be made in Figure 5 to identify the observations described in the article's text. To assess whether the carbon fibers promoted an increase in the thickness swelling property, it is important to have the density results of the panels. The authors did not present the dimensions or the percentage of raw material used to manufacture the panels. This information is necessary to verify whether adding FFCs caused the increase. The lack of statistical analysis also makes it impossible for the authors to state that there was a significant increase in properties caused by the addition of FFC fibers—a more in-depth discussion of the results obtained in the research needed to be included.
Citations from other researchers who justify the events, statements and information presented in the results must be included. Therefore, the authors should search the literature for recent citations to contribute to the results and observations presented.

Conclusions
Where can the developed materials be applied? Add the environmental, social and economic benefits the developed material can generate for the population and the industry. Present some suggestions for future studies.

Author Response

Dear editor and reviewers:

Thank you for your letter and the reviewers’ comments concerning our manuscript (Manuscript ID: polymers-3180395) entitled “Performance evaluation of carbon fiber fabrics-reinforced formaldehyde-free high-strength plywood”. Those comments are not only valuable and helpful for revising and improving our manuscript, also have the important guiding significance to our researches. We have studied the comments carefully and made relevant corrections which we hope to meet your approval. The main corrections in this manuscript and the responds to the reviewer’s comments are as following:

Reviewer 2:

Comments and Suggestions for Authors

Abstract

  1. (Line 28) - It's crucial to avoid repeating words from the title in the keywords. Let's replace the words: "high-strength" and "carbon fiber fabrics" to ensure the keywords are unique and effective.

Response: Thank you very much for the reminder. The corresponding words in the abstract have been corrected.

Introduction

  1. What is new about using this material? What benefits can it provide to industry, the environment and society?

Response: Thank you very much for this comment. In this study, maleic anhydride polyethylene (MAPE) polymer film was innovatively employed as a substitute for conventional formaldehyde-based adhesives. Concurrently, CFFs were incorporated as a reinforcing agent to fabricate high-performance plywood devoid of formaldehyde emissions. The obtained plywoods exhibit superior mechanical properties, formaldehyde-free, and intriguing thermoelectric conversion capabilities. The CFFs-reinforced plywood was benefit from simple, rapid, and feasible processing, which is expected to realize industrialized production. And all the raw materials used are sustainable and environmentally friendly, without complex chemical treatment. without chemical treatment. Therefore, a simple, sustainable, and processable CFFs-reinforced plywood would optimize the wood-based panel configuration and effectively enhance the social benefits.

Material and Methods

  1. Was the carbon fiber fabric used unidirectionally or bidirectionally?

Response: Thank you very much for this comment. We used carbon fiber fabric bidirectionally in this study. The corresponding contents have been added to the materials and methods of the revised manuscript.

  1. What is the thickness of the carbon fiber fabric?

Response: Thank you very much for this comment. The thickness of the carbon fiber fabric used in this study is 0.2 mm. The corresponding contents have been added to the materials and methods of the revised manuscript.

  1. What criteria were established for choosing the layer configurations between plywood and CFFs?

Response: Thank you very much for this comment. In this study, we explored the incorporation of CFFs as a reinforcement in plywood fabrication. According to the construction of plywood, CCFs can only be added between layers of veneer. However, we did not know in which layer it would be most appropriate to add CCFs, and therefore explored the effect of adding CCFs between all layers on the properties of the plywood. We designed the present experiment based on this consideration.

  1. What is the size of the panels produced? How many panels were manufactured for each treatment evaluated? How many samples were made for each physical and mechanical test performed? Why were density tests not performed on the materials? Density is a very important property for characterizing panels. I suggest adding it. The authors should include the physical and mechanical tests in different topics. Water absorption and Thickness swelling are physical and not mechanical analyses.

Response: Thank you very much for this comment. The original dimensions of the produced panels are 30 cm×30 cm. For each treatment, 3 panels were manufactured. And 5 samples were made for each physical and mechanical test performed, from which the mean value is derived for analysis. As the reviewer mentioned, Density is a very important property for characterizing panels. Therefore, the density of panels have been added to section 2.2 of revised manuscript. We have divided the mechanical and physical properties of the plates into two topics in the revised manuscript. Thank you very much for the reviewer's comment again.

  1. How were the samples prepared for the SEM? Were the samples metalized, and what material was used for the metallization? Provide more details on the analysis procedure.

Response: Thank you very much for this comment. In this study, the micromorphology of the dried CFFs-reinforced plywood (103 °C, 2h) was analyzed by scanning electron microscope (SEM). For better characterize the sample micro-morphology, all the SEM samples were metallized. Specifically, the SEM samples were cut into 1 × 1 × 0.5 cm3 blocks and sputtered with gold for observation. The experiment was performed by a Hitachi S-4800 SEM (Hitachi, Ltd., Tokyo, Japan) with a secondary electron (SE) mode at an accelerating voltage of 5 kV. The corresponding contents have been added into section 2.3 of revised manuscript.

  1. Detail the electrothermal conversion test. How were the samples prepared? What settings were used in the analyses? Did the analyses follow any standard or methodology?

Response: Thank you very much for this comment. The CFFs-reinforced plywood has the potential to be developed into a novel electrothermal device because of the excellent electrical conductivity of carbon fiber. Therefore, it is very necessary to evaluate the electrothermal conversion properties of the samples. The schematic diagram of the wood-based composite as an electrothermal conversion device is demonstrated in Fig. R1. Firstly, the CFFs-reinforced plywood was processed into 5 × 5 × 1 cm2 blocks. Subsequently, connecting copper wires to carbon fiber layers of samples. Finally, the samples were directly connected to an adjustable direct current (DC) power. And then a Decagon KD2 PRO thermocouple was used to monitor the temperature change in real time. As we all know, carbon fibers possess electrical exceptional conductivity conductivity, which can comparable to that of metals (Zhang et al., Joule 2, 764–777, 2018; Liu et al., Joule 1, 563–575, 2017). According to Joule's Law, when current flows through a resistor, the electrical energy is converted into thermal energy, which can be represented by the following formula:

Where,  represents the generated thermal energy,  represents the applied voltage,  represents the resistance, and  stands for the working time.

The joule heating effect theory shows that the thermal energy transformation of the wood-based composites has a positive correlation with the square value of the applied voltage (U2) at different working time. This means that only a low voltage is required to rapidly heating in carbon fiber layers. For safety considerations, we conducted experiment within a low-voltage range to investigate their thermoelectric properties. Thank you very much for the reviewer's comment again.

c

Figure R1. Experimental schematic diagram of Joule heating.

  1. This is important information that needs to be added. Why was no statistical test applied to the results obtained? It is important to check whether or not there is significance between the treatments evaluated.

Response: Thank you very much for this comment. The statistical test of obtained data have been added in revised manuscript. Thank you very much for the reviewer's comment again.

Results and Discussion

  1. To determine whether the values ​​indicated by the results showed significant differences between them, normalization, homogeneity, and variance analysis tests, followed by some mean test that applies to the analyses performed, must be performed. The fact that the values ​​are higher or lower is not enough to state that a significant difference occurred. For this, statistical tests are necessary. I suggest applying statistical tests to the results obtained.

Response: Thank you very much for this comment. As the reviewer mentioned, the statistical tests for the obtained data values are necessary. The one-way analysis of variance of obtained data have been added in revised manuscript. Thank you very much for the reviewer's comment again.

  1. (Line 176) – Figure 5 or Figure 2? Perform correction.

Response: Thank you very much for this suggestion. The error in the section 3.1 (line 176) has been corrected.

  1. Add scale to Figures 2c, 2d, 2e and 2f. What are the modulus of elasticity values during the tensile strength tests? Why did the authors not choose to display the results based on the stress x deformation curve? Standardize all results to two decimal places after the point. Add scale to Figures 3c, 3d, 3e and 3f. In Figure 4, does each image represent an analyzed material? If so, it is important to identify them in the Figure. Are the images in Figure 4 simulations of the analyzed material? What software was used? How was the material assigned? What are the characteristics of the mesh created for the analyzed materials? What failure criterion was used? The experimental procedures must be presented in the methodology if a simulation is performed.

Response: Thank you very much for this suggestion. The scales have been added to Figures 2 c-f, and Figures 3 c-f of revised manuscript. To test the strain of the plywood, we added an inducer to test the tensile strength, but the strain of the plywood was very small and beyond the test range of inducer, so we could not get the correct result. Therefore, the stress-strain curve is not recorded and the tensile modulus cannot be obtained. And all results have been standardized to two decimal places after the point.

The simulations of mechanical properties in Figure 4 have been presented in the experimental procedures.

  1. Markings should be made in Figure 5 to identify the observations described in the article's text. To assess whether the carbon fibers promoted an increase in the thickness swelling property, it is important to have the density results of the panels. The authors did not present the dimensions or the percentage of raw material used to manufacture the panels. This information is necessary to verify whether adding FFCs caused the increase. The lack of statistical analysis also makes it impossible for the authors to state that there was a significant increase in properties caused by the addition of FFC fibers—a more in-depth discussion of the results obtained in the research needed to be included.

Response: Thank you very much for this suggestion. The corresponding markings have been added to Figure 5 in revised manuscript. The density results of the panels have been added to section 2.2 in revised manuscript. The detail dimensions information of raw material have been added to section 2.1 in revised manuscript. The one-way analysis of variance of obtained data have been added in revised manuscript, and the results and discussion have been revised. Thank you very much for the reviewer's comment again.

14.Citations from other researchers who justify the events, statements and information presented in the results must be included. Therefore, the authors should search the literature for recent citations to contribute to the results and observations presented.
Response: Thank you very much for this suggestion. It is not only helps to improved our manuscript, but also broadens our knowledge. The corresponding literatures in results and discussion section have been added into the revised manuscript now. Thank you very much for the reviewer's comment again.

Conclusions

  1. Where can the developed materials be applied? Add the environmental, social and economic benefits the developed material can generate for the population and the industry. Present some suggestions for future studies.

Response: Thank you very much for this comment. The suggestions mentioned by the reviewers have been added to the conclusions section of the revised manuscript.

Round 2

Reviewer 1 Report

Comments and Suggestions for Authors

The given comments are answered.

Author Response

Dear editor and reviewers:

Thank you for your letter and the reviewers’ comments concerning our manuscript (Manuscript ID: polymers-3180395) entitled “Performance evaluation of carbon fiber fabrics-reinforced formaldehyde-free high-strength plywood”. Those comments are not only valuable and helpful for revising and improving our manuscript, also have the important guiding significance to our researches. We have studied the comments carefully and made relevant corrections which we hope to meet your approval. The main corrections in this manuscript and the responds to the reviewer’s comments are as following:

Reviewer 1:

  1. The given comments are answered.

Response: Thank you very much for the reviewer's recognition of our manuscript. Your suggestion not only helps us to correct the mistakes in our manuscript, but also expanded the depth of our research. Your recognition provides us with tremendous motivation to continue our further research.

Reviewer 2 Report

Comments and Suggestions for Authors

The authors have addressed most of the requested revisions. The article has been significantly improved.

Author Response

Dear editor and reviewers:

Thank you for your letter and the reviewers’ comments concerning our manuscript (Manuscript ID: polymers-3180395) entitled “Performance evaluation of carbon fiber fabrics-reinforced formaldehyde-free high-strength plywood”. Those comments are not only valuable and helpful for revising and improving our manuscript, also have the important guiding significance to our researches. We have studied the comments carefully and made relevant corrections which we hope to meet your approval. The main corrections in this manuscript and the responds to the reviewer’s comments are as following:

Reviewer 2:

  1. The authors have addressed most of the requested revisions. The article has been significantly improved.

Response: Thank you very much for the reviewer's recognition of our manuscript. We sincerely hope that this revised manuscript has addressed all your comments and suggestions. We appreciated for reviewers’ warm work earnestly, and hope that the correction will meet with approval. Please do not hesitate to contact us if there are any question. Thanks again to the reviewers and editors for your hard work! Best wishes to you!
